# A Multifunctional Battery-Free Bluetooth Low Energy Wireless Sensor Node Remotely Powered by Electromagnetic Wireless Power Transfer in Far-Field

**DOI:** 10.3390/s22114054

**Published:** 2022-05-27

**Authors:** Alassane Sidibe, Gaël Loubet, Alexandru Takacs, Daniela Dragomirescu

**Affiliations:** Laboratoire d’Analyse et d’Architecture des Systèmes du Centre National de la Recherche Scientifique (LAAS-CNRS), Université de Toulouse, Centre National de la Recherche Scientifique (CNRS), Institut National des Sciences Appliqués de Toulouse (INSA) de Toulouse, Université Paul Sabatier, Toulouse III (UPS), 31400 Toulouse, France; gael.loubet@laas.fr (G.L.); alexandru.takacs@laas.fr (A.T.); daniela.dragomirescu@laas.fr (D.D.)

**Keywords:** wireless power transmission (WPT), wireless communications, wireless sensor node, Bluetooth low energy (BLE), Internet of Things (IoT), power management unit (PMU), structural health monitoring (SHM)

## Abstract

This paper presents a multifunctional battery-free wireless sensing node (SN) designed to monitor physical parameters (e.g., temperature, humidity and resistivity) of reinforced concrete. The SN, which is intended to be embedded into a concrete cavity, is autonomous and can be wirelessly powered thanks to the wireless power transmission technique. Once enough energy is stored in a capacitor, the active components (sensor and transceiver) are supplied with the harvested power. The data from the sensor are then wirelessly transmitted via the Bluetooth Low Energy (BLE) technology in broadcasting mode to a device configured as an observer. The feature of energy harvesting (EH) is achieved thanks to an RF-to-DC converter (a rectifier) optimized for a low power input level. It is based on a voltage doubler topology with SMS7630-005LF Schottky diode optimized at −15 dBm input power and a load of 10 kΩ. The harvested DC power is then managed and boosted by a power management unit (PMU). The proposed system has the advantage of presenting two different power management units (PMUs) and two rectifiers working in different European Industrial, Scientific and Medical (ISM) frequency bands (868 MHz and 2.45 GHz) depending on the available power density. The PMU interfaces a storage capacitor to store the harvested power and then power the active components of the sensing node. The low power digital sensor HD2080 is selected to provide accurate humidity and temperature measurements. Resistivity measurement (not reported in this paper) can also be achieved through a current injection on the concrete probes. For wireless communications, the QN9080 system-on-chip (SoC) was chosen as a BLE transceiver thanks to its attractive features: a small package size and extremely low power consumption. For low power consumption, the SN is configured in broadcasting mode. The measured power consumption of the SN in a deep-sleep mode is 946 µJ for four advertising events (spaced at 250 ms maximum) after the functioning of sensors. It also includes voltage offset cancelling functionality for resistivity measurement. Far-field measurement operated in an anechoic chamber with the most efficient PMU (AEM30940) gives a first charging time of 48 s (with an empty capacitor) and recharge duration of 27 s for a complete measurement and data transmission cycle.

## 1. Introduction

In the field of the Internet of Things (IoT), small-size, multifunctional and ultra-low-power systems are required to address issues of miniaturization, manufacturing, deployment and maintenance costs. Reducing power consumption is one of the main challenges when designing these systems. Thus, the need to continuously power electronic devices during operation has motivated the development of an energy harvesting technique. Moreover, in the frame of the deployment of multiple connected and communicating objects for long-term applications, the conventional power supply methods show their limitations: batteries need to be changed periodically and recharged, and capacitors and supercapacitors have limited energy storage capacities. Based on such limitations, a low-power and low-cost IoT device for field data gathering in precision agriculture practices was proposed in which the battery was charged with a solar panel [1].

The aim of this work is to monitor the physical parameters of a reinforced concrete beam using an embedded wireless sensor. It should be autonomous and able to transmit from inside the concrete to the communication nodes placed at a certain distance. A detailed explanation of its purpose and architecture is given in Section 2.

In this paper, we will present the architecture of the proposed system, which consists of the following: a rectifier, specifically designed to convert the radiofrequency (RF) power to DC power energy and supply a wireless SN; two power management units (PMUs) to efficiently manage and store the required energy in a storage capacitor; sensors and a transceiver that allow wireless transmission of the measured temperature and humidity data. The SN also offers the capability of harvesting EM waves at different frequencies. Both rectifiers were tuned to work in the European Industrial, Scientific and Medical (ISM) 868 MHz and 2.45 GHz frequency bands. Their design and performance will be presented. Finally, a comparative test between both PMUs and results of the use of the complete system was performed in an anechoic chamber with the most efficient PMU.

## 2. Targeted Application: Structural Health Monitoring

The concept of “communicating material” is presented as a new paradigm for industrial information systems firstly introduced in [2]. The material will be able to measure, process, store and communicate data thanks to embedded wireless sensor devices meeting the requirements of the IoT paradigm. Advanced studies showed interest in this concept for materials such as wood and textiles [2,3]. The capability of wireless communication in reinforced concrete was also studied through an embedded sensor for permeability experiments in [4] and temperature monitoring through an embedded sensor tag in [5]. The works achieved in this field have proposed interesting results by providing diverse functionalities to the users all along the lifecycle of the materials, as explained in [6,7]. The previous version of this work, which is dedicated to monitoring the physical state of reinforced concretes, was implemented with a LoRaWAN technology [8]. To achieve such an objective, research activities must face several obstacles when sensing nodes (SNs) are embedded into concrete (currently in air cavities):-The need for reliable and robust wireless communication that enables the signal to be received from and through the materials, regardless of their composition (e.g., reinforcements) and state (e.g., wet, or dry);-The need for energy autonomy despite the physical inaccessibility of the sensing nodes (e.g., for replacing their batteries)-The choice of a secure, long-range and trusted wireless communication between sensing nodes and of a data management strategy controlling how data are spread.

Specific solutions should be found for this kind of application where the size and the lifespan of the sensing nodes are limiting factors.

### 2.1. Lifetime Issues: Wireless Power Transfer

While working in the electronic domain, we are most of the time faced with power issues. Meanwhile, electronic devices are becoming more and more powerful with multiple and outstanding functionalities, thus always requiring more power. A WSN for temperature and humidity monitoring within concrete structures was previously presented in [9]. However, this solution is battery-operated, which will limit its service lifetime.

Many research activities were completed to demonstrate the feasibility of using ambient energy harvesting (EH) to power battery-free devices efficiently from ambient energy sources. The most widely utilized ambient energy sources are summarized in [10] (pp. 1650–1651). Solar power is commonly used, but it is unpredictable as the available power is a function of the climate, the size and the orientation of the solar panel and the time of the day and of the year. Thermal energy harvesting is an alternative to the solar EH to power wireless sensors, as shown in [11,12]. It works by converting the temperature gradient into electric power (the Seebeck effect) thanks to a thermoelectric generator (TEG) [13] but has a limited power conversion efficiency. Based on using mechanical deformation to generate electric power, the piezoelectric EH has found interest in wearable and healthcare applications [14,15]. Finally, the radiofrequency (RF) EH solution consists of converting ambient electromagnetic waves to DC power. However, EH technique does not allow sufficient harvested power due to the limited power density level from ambient EM waves (GSM, Bluetooth, WiFi and GPS) compared to solar energy. The available power density obtained on in RF survey performed in [16], is less than −25 dBm/cm^2^ for several frequency bands. An alternative to harvest EM waves from a dedicated RF energy source is possible thanks to wireless power transfer (WPT) or wireless energy transmission (WET). Two categories of WPT exist, and each presents advantages and disadvantages suitable for a specific application. The near-field (NF) WPT is mostly used for high-power devices or systems with a close distance (on the order of centimeters) between the power transmitter and the power receiver [17]. Unlike NF, far-field (FF) WPT allows spreading EM waves from a dedicated RF power source through an transmitting antenna at a certain distance from the receiver [18]. The FF distance is limited by the frequency and the transmitter and receiver antenna gain. The Table 1 summarizes the advanced research on the state of the art using different energy sources for adequate applications.

Regarding the advantages and disadvantages of each EH solution, the most suitable to be used for wireless sensors embedded in an air cavity or a reinforced concrete cavity is the RF EH or WPT solution. Recent progress in the field of WPT for wirelessly powered sensor networks is presented [22]. It has proved its efficiency through some developments reported in [23,24].

### 2.2. Wireless Communication: Bluetooth Low Energy

A reliable and secure communicating protocol must be used for the implementation of a WSN for SHM applications. It must also present the advantage of transmitting a signal that can be received at a long range through reinforced concretes. Relevant characteristics of the wireless communication technologies are reported in a table in [25] (pp. 30–33). An interesting passive RFID wireless sensor system for moisture monitoring in concrete, which is supplied with power exclusively by the electromagnetic field from an external reader device, was developed in [26]. Despite being battery-free and fully passive, this solution presents the disadvantage of using the near-field communication (NFC) frequency band of 13.56 MHz and thus has a limited reading range of 3 cm. An RFID-based wireless system with a central frequency of 868 MHz for temperature monitoring of concrete is presented in [27]. The analytical results estimated a maximum distance of 1 m from the reader to the RFID sensor embedded at 0.15 m inside concrete. The challenges of using wireless RFID sensor are studied in [28]. The main limitations reported are related to a limited EH and read range, the sensor responses collisions, the cost of the reader and the lack of UHF RFID mobile sensing platforms.

A LoRaWAN protocol was previously chosen for the implemented solution thanks to its advantages (low cost and very long-range communication) [8]. It demonstrated the feasibility of monitoring, processing and storing the physical parameters of the concrete with a fully embedded sensing mode. The experimental results allow a communication range of 1.3 km from the sensing node to a LoRA gateway. In this work, a trade-off has been made between the range and the power consumption by choosing BLE technology. The BLE technology has a higher data rate than LoRaWAN; thus, it is possible to send a large amount of data at once. Further, it can be possible to receive and have access to the monitored data with any wireless devices using BLE technology (such as smartphones, tablets and even smartwatches).

#### 2.2.1. Characteristics of the Bluetooth Low Energy

The BLE wireless communication technology is designed for low-power operation. It operates in the 2.4 GHz Industrial, Scientific and Medical (ISM) frequency band from 2.402 GHz to 2.48 GHz [29]. The covered channels are separated into two parts with 3 advertising channels and 37 data channels. Related to the used type of connection (point to point, mesh or broadcast), BLE can operate in four different modes (broadcaster, observer, central or peripheral).

For optimized power consumption and knowing that the targeted application does not require point-to-point or mesh communication, a simplified broadcaster/observer mode is configured. The broadcaster device is used to transfer the data in the advertising packets on the three primary advertising channels without any incoming connection. The broadcast event corresponds to the duration of the primary advertising channels 37, 38 and 39. These channels are respectively transmitted at the frequencies of 2.402 GHz, 2.426 GHz and 2.480 GHz as seen in Figure 1.

A payload sets each adverting packet with at least 37 bytes containing the advertising specified data (advData that are the unique identification and the data measured by sensors). The observer will continuously scan, in passive mode, the advertisements on the three dedicated channels from other SNs. In addition, for accurate data transmission, both devices, broadcaster and observer, are synchronized by configuring the same advertising interval.

#### 2.2.2. BLE System-on-Chip (SoC)

Besides the interesting features of the BLE protocol, the choice of the components is critical when designing a system with the lowest energy consumption. Several BLE transceivers are now available. A discussion on the choice of the right BLE SoC is important before sizing the global device. Most of these devices are compared and reported in Table 2 as a function of the key relevant features. It can be observed that most of the commercialized transceivers have roughly the same sensitivity level of around −95 dBm, which can enable a coverage range typically over several tens of meters. The default transmitting power is generally defined at 0 dBm, which is enabled for all transceivers. Regarding the receive (Rx) and transmit (Tx) current, the QN908x has the lowest value despite its high current consumption in deep-sleep mode compared to the RSL10 transceiver. The resolution of the analog-to-digital converter (ADC) is also an important parameter to consider, and most of the transceivers support at least 10 bits of resolution.

## 3. Architecture of and Design of the Proposed Wireless Sensing Node

The functionalities of the sensing node are implemented based on the proposed block diagram as seen in Figure 2. The proposed architecture is based on a previous version of the designed sensing node reported in [25] (pp. 213–227), which presents the same functionalities but with a unique PMU and unoptimized power consumption. It consists of the following subsystems: two single-band rectifiers allowing RF-to-DC conversion at different frequencies; a power management unit (PMU); a temperature and relative humidity sensor; a resistivity measurement circuit; and a BLE system-on-chip (SoC) transceiver. The SN is wirelessly powered by a rectenna, which is composed of the integrated onboard rectifier for low power input levels and an external antenna. The PMU is set up to manage, boost and store the available power from the rectifier. Then, it provides the needed power to the sensors and BLE SoC even in deep-sleep mode. This mode is considered when there is no available energy in the storage capacitor and all nodes are deeply discharged.

The resistivity sensing is not reported in the work. With the SN, it is possible to inject a current into the concrete to compute the electrical resistivity by measuring a potential difference thanks to probes embedded in the concrete. The evolution of this physical parameter will allow estimating the corrosion rate.

### 3.1. Design of the RF-To-DC Conversion Circuit: The Rectifier

The WPT technique is chosen to wirelessly power the implemented SN. It consists in generating and transmitting time-varying EM waves across space from a power source connected to an antenna to the receiving antenna of the SN. When the receiving and transmitting antennas operate at the same frequency and are perfectly positioned with the same polarization, enough power will be captured, and thus the RF power from the receiving antenna will be converted to DC power thanks to the rectifier subsystem.

In this work, we will focus on powering first the SN without an external antenna for a proof of concept. Therefore, the RF-to-DC converter (rectifier) embedded in the SN is presented. We have made the choice of using two separate single-band rectifiers at different frequencies rather than a dual-band rectifier to avoid not only the power dependence from one rectifier (since it is damaged) but also the use of two external antennas on the opposite side. By using a single-band rectifier, we also optimize the power conversion efficiency thanks to a better matching network. Research activities have studied the comparison between single-band and dual-band rectifiers. In [37], a rectifier at 2.45 GHz and a rectifier working at 2.45 GHz and 5.8 GHz are introduced. Based on the design without an impedance compression network, the conversion efficiency for the single-band rectifier is 10% less than that for the double band at the same frequency (2.45 GHz) with an input power of 0 dBm (which is not enough low to see the difference). This drop in efficiency can also be observed in [38] when comparing the single-band and dual-band rectifiers at 900 MHz. Our strategy in the implementation of the SN is to use the same rectifier design and select the working frequency by tuning the L-matching network. Different typologies, selected diodes and rectifier frequencies are presented in the state of the art [18,39]. Apart from the topology, the choice of a low-barrier Schottky diode is crucial in achieving better performances in terms of DC voltage and efficiency for weak input signals. A comparison of different types of Schottky diodes is proposed in [40,41]. It shows that the SMS7630 series from Skyworks is more efficient for RF input power lower than 0 dBm due to its low forward voltage of 240 mV. The SMS7630-005LF model was selected in this work [42]. The rectifiers are optimized to work at low input power and present the advantage of being compact. The voltage doubler topology increases the harvested DC voltage across the load (compared to the single-diode rectifier). However, in terms of efficiency, a single or half-wave rectifier is often preferred, mostly for very-low-power applications. Its schematic is presented in Figure 3. It consists of an LC matching network, a series pair Schottky diode (SMS7630-005LF) and an output capacitor (a part of the low-pass filter with load considered as the input impedance of the PMU). The component values and references are given in Table 3. The design, simulation and experimental results were detailed in previous work in [43]. Quick experimentations of the implemented SN give a DC voltage across a 10 kΩ load with a −15 dBm input powers of 286 mV and 236 mV for the 868 MHz and 2.45 GHz rectifiers, respectively.

### 3.2. Sensing Subsystem

This subsystem is dedicated to measuring the targeted parameters and formatting and wirelessly transmitting the collected data to the observer. It is powered by the DC power available from the PMU, which stores energy in the capacitor from the power available at the output of the rectifier. The BLE SoC and active sensors are then supplied when there is sufficient energy available in the capacitor. Depending on the rectifier chosen, the output can be selected using the jumper J1. In this work, we opt for using two commercialized PMUs. These are tested and characterized; the performances are then compared to determine the advantages of each according to the available electromagnetic power density and the input power required for the sensors to be used. An integrated temperature and humidity sensor HDC2080 with low power consumption is chosen from TI [44]. Depending on the use, the input (the DC voltage source) and the PMU can be selected by jumpers J1 and J2. The sensors and BLE transceiver can be also powered from an external DC source.

#### 3.2.1. Bluetooth Low Energy Transceiver QN9080

Wireless data communication is enabled by an ultra-low-power BLE, the QN9080 SoC [30] and a conventional Meandered Inverted-F Antenna (MIFA) designed at 2.45 GHz as suggested in the application note [45].

##### Algorithm and Configuration

The BLE transceiver is programmed to operate only as a broadcaster, sending the measured temperature and humidity data in advertising mode. A paper has examined packet collisions for the BLE advertising mode [46]. The probability of packet collisions can be decreased by increasing the advertising time and the number of packets, but this increases the amount of energy required. We found a trade-off by sending four advertising packets over the three allocated channels (37, 38 and 39) with an advertising interval of 250 ms. The packets are sent to the observer which is a QN9080 NPX development board configured in observer mode [47].

As depicted in Figure 4, the broadcasting procedure starts with a device initialization which takes into account the initialization of variables, the central processor unit (CPU), peripherals and the host stack of the BLE protocol. The initConfig_sensor function is dedicated to initializing the configurations of the HDC2080 sensor (about all registers, GPIO and I^2^C pins and the timestamp module). The following step consists of triggering the measurements through an I^2^C command by writing 0x01 on the allocated address of the register. The trigger on demand mode allows the device to remain in sleep mode when it is not requested. At the same time, the Voltage_Meas Resistivity task, which provides resistivity measurement (not reported in this paper), is run before the sensor operation for accurate measurement when the voltage is stable.

Thereafter, the measurement can start with the DataMeas_HDC2080 task where the writing on the register function is called back before the data value on the register is read. The returned value (Data_Valid) from this function is checked to validate or not the proper functioning of the measurement. If “Data_Valid” is true, the relative humidity (RH) and temperature (T) are converted; otherwise, the measurement process end. The integer and the decimal parts of the measured data are allocated on 7-bit for a temperature range from −40 °C to +85 °C and 2-bit for an accuracy of 0.25 °C, respectively. The data conversion is followed by the Generic Access Profile (GAP) configuration of the BLE stack in broadcasting mode and the setting of the advertising parameters. A timer of 1 s is activated to allow four advertising events with an interval of 250 ms. When the timer ends and the four data are sent, the controller stops the advertising. Finally, it is mandatory to force the discharge of the storage capacitor to enable the PMU to complete another charging procedure from the deactivation threshold (detailed in the next section) and thus complete another data measurement and advertising step.

##### Evaluation of the Power Consumption

To configure the PMUs with the provided configuration tools from the manufacturer, the total power required by the sensors and transceiver has been evaluated. This can be carried out by determining the power budget (sum of the consumed power at each stage) or using the power measurement tool in the NXP software (MCUXpresso IDE v11.1.1 [Build 3241] [2 March 2020]) [48].

The current consumption profile of the SN during broadcasting with a DC voltage of 3 V is represented in Figure 5. It goes through the following steps:(1)Without any available or sufficient power, the SN stays permanently in power-off mode without any activity of the hardware.(2)The functioning of the SN starts with an inrush current followed by register initialization and calibration of the sensors. This stage represents a large part of the consumption during the broadcasting phase. The SN has a high peak demand of 14.6 mA in the start-up process once powered by DC voltage.(3)A following adverting event after the initialization phase is produced to avoid restarting the MCU, and thus a supplementary consumption as seen in the next three advertising events. The advertising interval is set to 250 ms and a timer of 1 s is implemented to send four advertisements. Each advertising event starts with a wake-up of the MCU and transmission of packets on the dedicated channel (37, 38, 39). A detailed view of advertising packets is shown in Figure 6.(4)After each advertising event, the SN goes into sleep mode with low current consumption. The average current measured is 27 µA.(5)The last current consumption phase is produced by the function to stop advertisement after the timer of 1 s. An additional function is programmed for the resistivity measurement by canceling the measured current across the probes.

The total average current consumption during a broadcasting phase is approximately 282 µA during the 1216 ms. It can be computed thanks to Formula (1) but is less accurate due to the approximation of time duration at each state.
(1)$$I_{average}=\frac{\sum^{\;}I_{i}T_{i}}{\sum^{\;}T_{i}}=\frac{I_{2}\cdot T_{2}+4\cdot I_{1}\cdot T_{1}+3I_{3}T_{3}+I_{4}T_{4}}{4\cdot T_{1}+T_{2}+3\cdot T_{3}+T_{4}}$$

The energy consumption of the SN is then 1028.7 µJ (E_cons_) with a DC voltage of 3 V. However, this method of calculating power consumption is not accurate due to the variation in offset voltage, which induces an error for measured currents below 150 µA (as described in the datasheet [49]). An alternative was found by increasing the advertising interval and measuring the voltage across a resistor with a digital multimeter (Keithley 2000). The resulting current is 3.3 µA instead of 31.6 µA in a deep-sleep state as reported in Table 4. According to Formula (1), the correct value of the average current is 260 µA, resulting in the energy consumption of 946 µJ (E_cons_).

Thus, the final storage capacitor is determined thanks to Formula (2) for both PMUs by defining the maximum (activation) and minimum (deactivation) threshold voltage at 4.2 V and 2 V, respectively, and subtracting 20% from the required value to compensate the tolerance of the capacitance value in the worst case ($C_{stor-20\%})$. The value above is chosen knowing that the calculated value of 111 µA is not standard. The low-leakage-current storage capacitor of 150 µF [50] from Würth Elektronik is selected for the prototype.
(2)$$E_{cons}=\frac{1}{2}\cdot C_{stor-20\%}\cdot \left(V_{max}^{2}-V_{min}^{2}\right)$$

#### 3.2.2. The BQ22570 Power Management Unit

Knowing that the instantaneously available power from the harvester is not sufficient to supply the BLE transceiver and active sensors continuously, we decided to manage, boost and store it in a storage capacitor of 150 µF previously calculated, with the BQ25570 PMU as a low-power device (its minimum input DC power is around 15 µW for 100 µF) [51]. Thanks to the configuration tools, we obtain an activation voltage of 4.195 V and a deactivation voltage of 1.936 V with an output voltage of 2.87 V. Figure 7 represents the evolution of the DC voltage across the storage capacitor for an RF power of −6 dBm (868 MHz) at the input of the rectifier. The real amount of energy stored in the capacitor is 1.04 mJ.

As seen in Figure 7, from an empty storage capacitor, the PMU requires a long cold start procedure and recharge of 114 s and 21 s, respectively, for an RF power of −6 dBm at the input of the rectifier. The recharge time is then shorter while the PMU allows activation of the maximum power point tracking (MPPT) hardware system.

#### 3.2.3. The AEM30940 Power Management Unit

This chip from e-peas [52] is a full-featured energy-efficient PMU that can be used to generate the appropriate regulated supply voltage. It was chosen because of its ultra-low-power start-up with a typical input power of 3 µW and can be configured to store the energy for different elements (supercapacitor or conventional capacitor, thin-film battery, Li-ion battery, LiFePO4 battery, etc.). The low drop-out voltage can generate various supply voltages: low voltage (LV) supply (1.2 V to 1.8 V) for microcontrollers and high voltage (HV) supply (1.8 V to 3.3 V) for transceivers. An additional voltage path for a rechargeable battery is possible with the DC-DC boost converter (2.2 V to 4.5 V). The HV LDO, which consumes less than the boost converter, is selected for our proposal. With its configuration tool, the threshold voltages are obtained for 4.22 V and 1.87 V, and the output high voltage of the LDO is set to 2.87 V.

Thanks to its ultra-low-power start-up, the cold start procedure takes only 13 s, but the recharge time is longer (around 21 s) as seen in Figure 8. The next section will detail the experimental results of the prototype with the same configuration as described above.

## 4. Implementation and Experimental Results

The proposed solution of the SN combining two PMUs and two rectifiers at different frequency bands was prototyped. The nomenclature of each subsystem of the proposed SN is presented in a photo (Figure 9). In this chapter, we will present the implementation of both PMUs. The output of either the 2.45 GHz rectifier or the 868 MHz rectifier input can be used for the power supply, but only the experimental results of the SN with the 868 MHz rectifier are presented for the first proposal of this paper.

### 4.1. Comparison of the Power Management Unit

The performance was evaluated first by measuring the charge time of the storage capacitor and second by comparing each PMU. Experimental results were obtained by providing an RF input signal at variable power levels to the 868 MHz rectifier, and the charge and discharge evolution could be obtained thanks to an oscilloscope, as depicted in the experimental setup in Figure 10. To avoid the influence of the input impedance of measuring devices (voltmeter, probes and oscilloscope), the time duration of a single emission or the time between data emissions could be obtained by determining the time difference between each received timestamped packet.

The voltage evolution in time is reported in Figure 11 for: V_cc_, the supply voltage to the transceiver and sensors; V_cstor_, the voltage across the empty storage capacitor; V_DCin_, the input voltage of the PMU from the output of the rectifier. It can be clearly observed that the configuration with BQ25570 requires a long cold start time, while the configuration with AEM30940 completes the cold start procedure and allows four advertising events. We will therefore say that the AEM30940 seems to be more suitable for our application.

A quantitative comparison has been made between the two PMUs related to the first charge and recharge duration as a function of the input power level (in Figure 12). The first charge duration includes the cold start time. The evolution of the first charge duration meets the insignificant dynamic range for the two PMUs independently of the input power. On the other hand, the recharge duration is roughly the same for input power higher than −8.5 dBm, but the difference between the PMU increases for low RF input power. As a result, the AEM3090 PMU offers interesting results with reasonable charge duration, especially for low power input levels.

### 4.2. Radiated Performance Evaluation

In order to be finally implemented in real wireless conditions by connecting the SN with a compact 3D configured antenna [53], the antenna was optimized to have a maximum gain higher than +1 dBi and a size respecting a planar dimension of 60 mm × 30 mm. Thanks to miniaturizing techniques consisting of adding two vertical metallic arms on the edges, the obtained antenna has a size of 56 mm × 32 mm × 10 mm. It has a maximum measured gain of +1.54 dBi at 868 MHz and can be used in the bandwidth between 862 MHz and 888 MHz. The full prototype is shown in Figure 13. In this characterization step, the SN is configured only for AEM30940, knowing that it is more suitable for low RF power levels as determined before.

The setup represented in Figure 14 is composed of: an RF source which consists of an RF signal generator (MG3690C) connected to a patch antenna through a coaxial cable; the SN considered as the broadcaster under test placed at a distance of 2 m from the RF source; and the observer which is the QN9080 Development Kit that enables the transmitted data to be received by the SN.

The used patch antenna operates at 868 MHz and has a maximum gain of +9.4 dBi. According to the configured output power from the signal generator, the equivalent received signal at the input port of the rectifier can be computed thanks to Formula (3). Before any measurements, the losses induced by the cable were measured thanks to a power meter. Thus, Figure 15 shows the first start duration and the recharge periodicity of the storage capacitor as a function of the effective isotropic radiated power.

The measurement and data transmission periodicity can be controlled by the RF power source through the equivalent power density level at the surface of the antenna connected to the SN.

## 5. Conclusions

Faced with the ambition of extending the lifespan of electronic systems, especially in harsh environments for SHM applications, this paper presents a low-power, autonomous and multifunctional sensing node dedicated to wirelessly communicating, via BLE protocol, the data of the physical parameters (temperature, pressure and resistivity) measured by sensors. Thanks to its low power consumption, the BLE protocol configured in broadcaster/observer mode was chosen instead of LoRaWAN previously used in [8]. The SN used as a broadcaster simply sends out data to the observer which periodically scans the data without any connection.

The SN has the advantage that it can be configured according to the power density level. A rectifier optimized for −15 dBm at an 868 MHz frequency band is used as an RF-to-DC converter to supply the active components. The SN is designed and implemented with both PMUs (BQ25570 and AEM30940); experimental results show a lower charge duration of the AEM30940 for the cold start independently of the input power level. For an input power of −6 dBm at 868 MHz at the rectifier input, the cold start duration of the BQ25570 is 4 times longer. However, the trend is reversed after the first charging step, where the BQ2557 recharges the storage capacitor faster. Far-field experimentation of the SN was also completed in an anechoic chamber at 2 m from the ES at 868 MHz. With an EIRP lower than +27 dBm of an equivalent power density, the SN can complete a full first charge in less than 4 min and has a recharge periodicity of less than 2 min.

This paper presents the performance results of the developed battery-free sensor node powered by a far-field RF power transmission technique. With the advantages of being low-power and battery-free and enabling wireless BLE communication, the SN is intended to be embedded into a concrete cavity for structural health monitoring. The next step in this line of research will be to carry out experiments on the SN with both frequency bands (2.45 GHz and 868 MHz). Moreover, this prototype will be optimized by reducing the size of the SN board and by co-designing an antenna on the same printed circuit board (PCB).

## Figures and Tables

**Figure 1 sensors-22-04054-f001:**
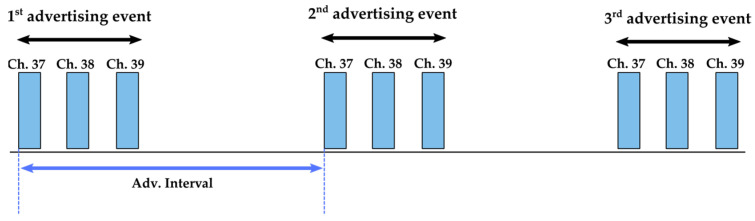
Legacy advertising.

**Figure 2 sensors-22-04054-f002:**
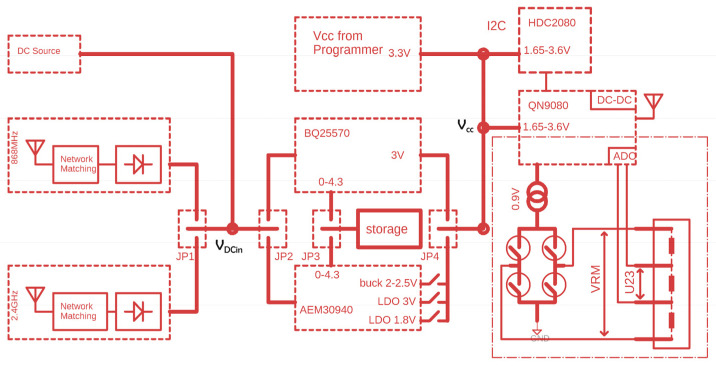
Block diagram of the architecture of the implemented sensing node.

**Figure 3 sensors-22-04054-f003:**
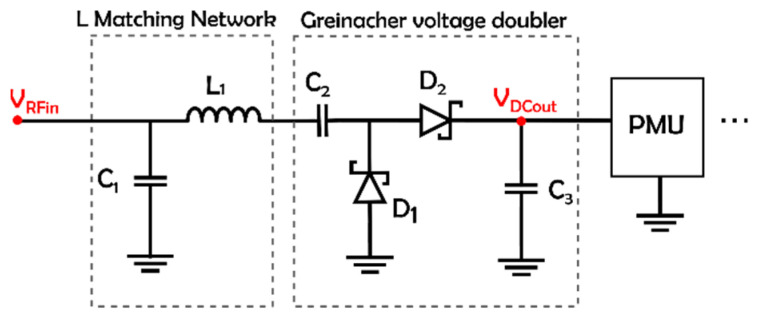
Schematic of the implemented rectifier.

**Figure 4 sensors-22-04054-f004:**
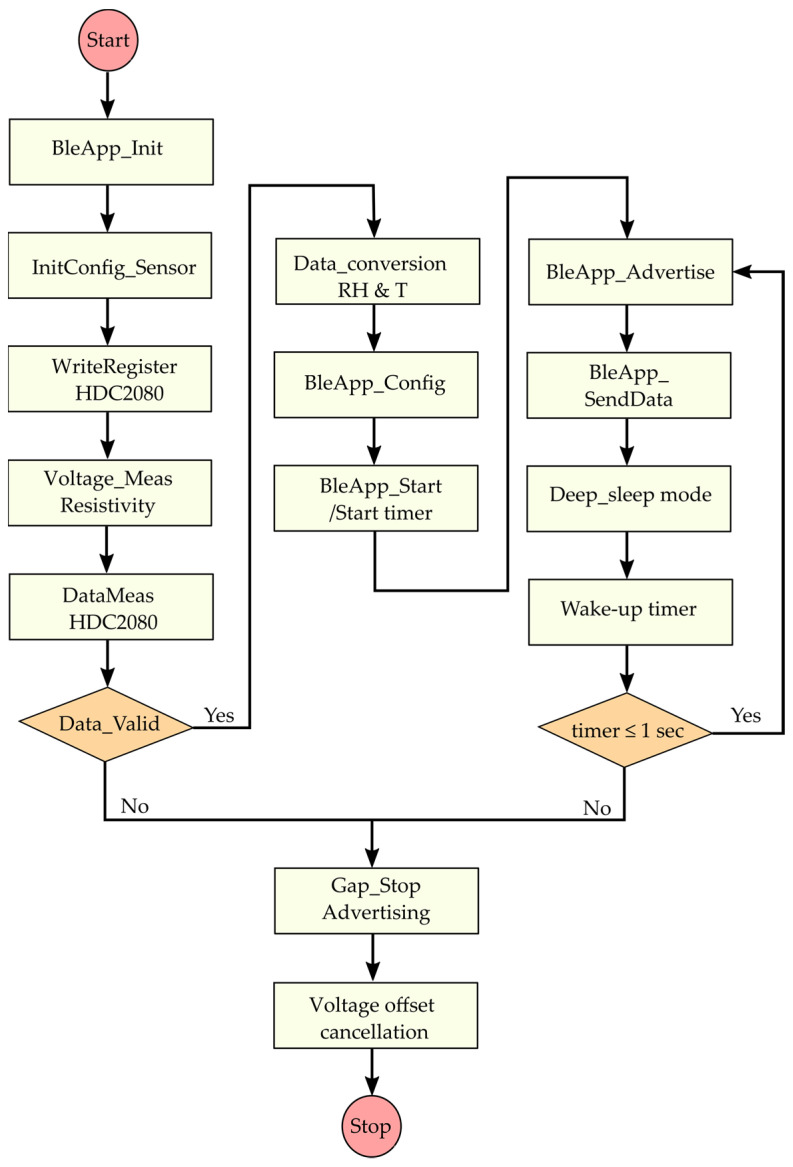
Flowchart of the functioning of the BLE SN in broadcasting mode.

**Figure 5 sensors-22-04054-f005:**
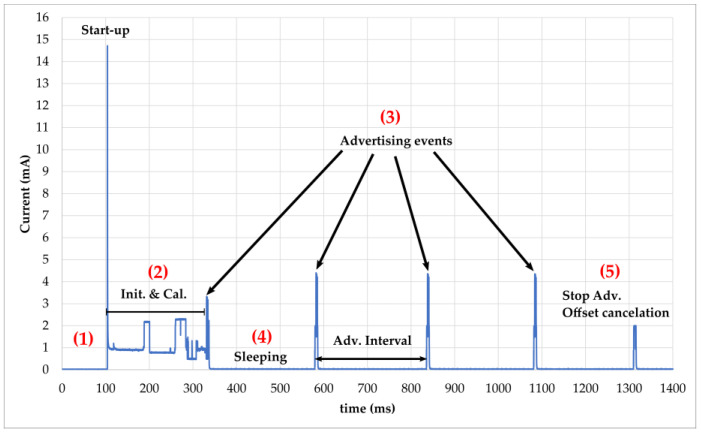
The current consumption profile of the BLE SN during broadcasting, obtained with the power measurement tool of MCUXpresso IDE software. The different states are identified: (1) SN is power off; (2) Indicates the start-up, initialisation and calibration after power supplying the SN; (3) is the 4 advertisements made by the SN with data; (4) is the sleep mode between adv. event for low power consumption; (5) is the consumption needed to stop broadcasting and cancel the voltage offset for resistivity measurement.

**Figure 6 sensors-22-04054-f006:**
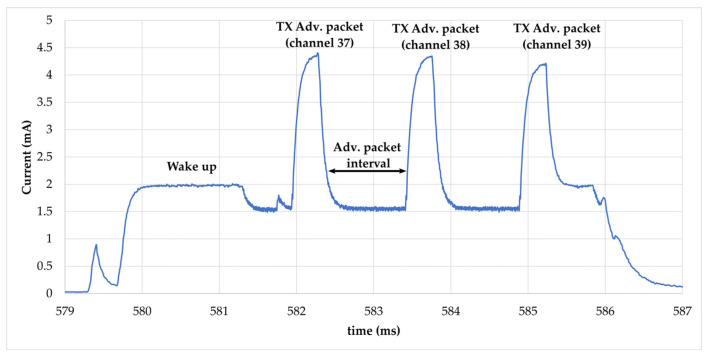
Current consumption profile of the BLE SN during advertising event.

**Figure 7 sensors-22-04054-f007:**
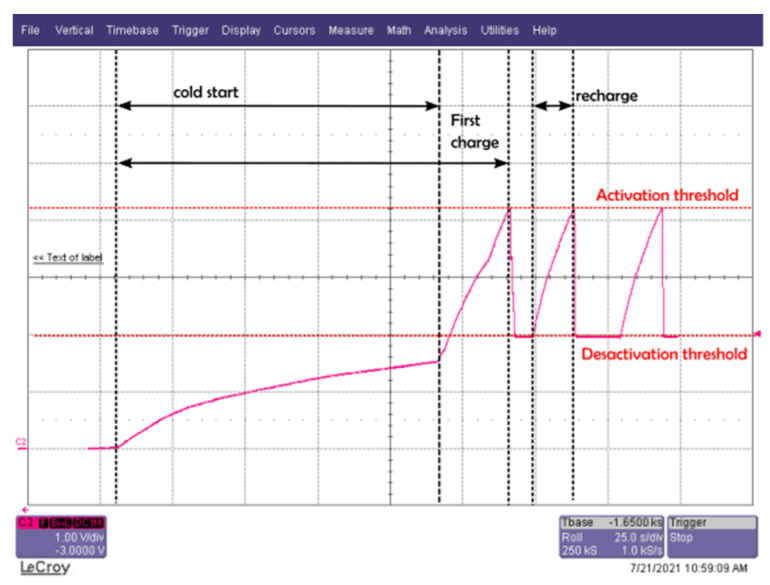
Load time of the storage capacitor (cold start, first recharge and then a recharge) with the BQ25570 for an RF power of −6 dBm (868 MHz) at the input of the rectifier.

**Figure 8 sensors-22-04054-f008:**
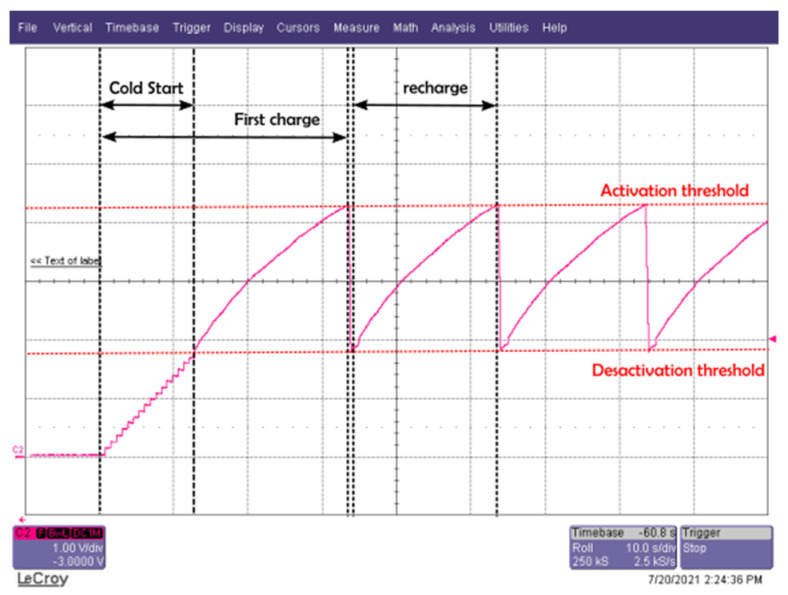
Load time across the storage capacitor (cold start, first recharge and then a recharge) with the AEM30940 for an RF power of −6 dBm (868 MHz) at the input of the rectifier.

**Figure 9 sensors-22-04054-f009:**
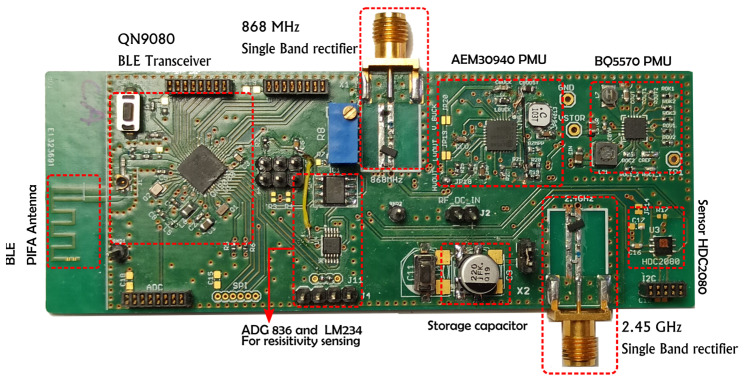
Photo of the fabricated sensing node with the nomenclature of each part.

**Figure 10 sensors-22-04054-f010:**
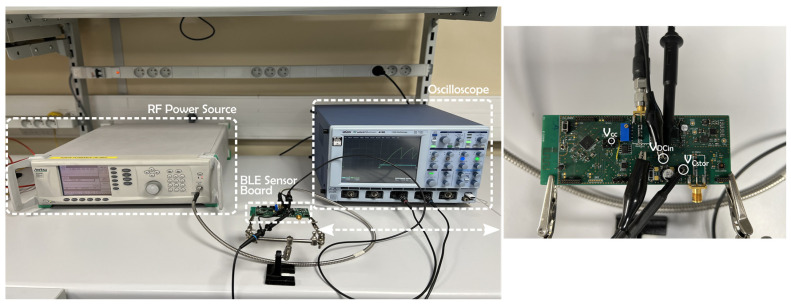
Experimental setup of measuring the charging evolution of the storage capacitor.

**Figure 11 sensors-22-04054-f011:**
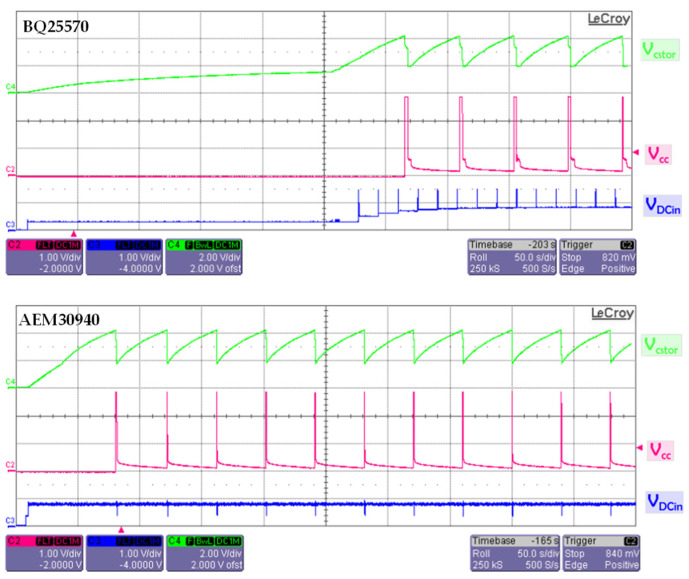
Voltage evolution in time from an empty storage capacitor of the configuration with PMU BQ25570 (**top**) and AEM30940 (**bottom**).

**Figure 12 sensors-22-04054-f012:**
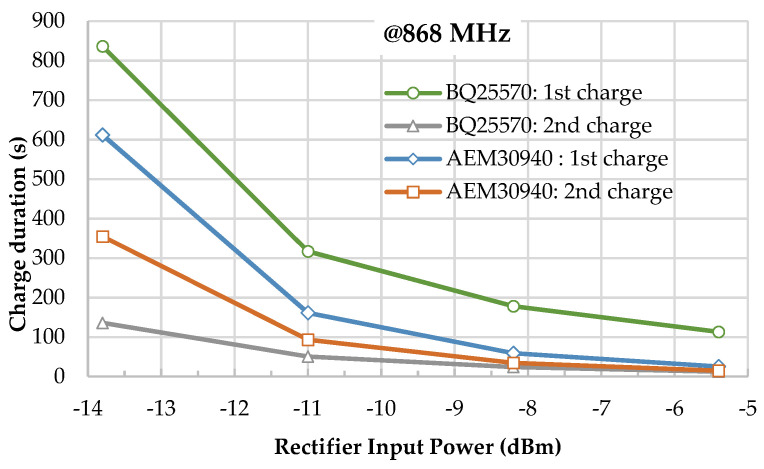
Charge duration of the SN with an 868 MHz rectifier for different selected PMUs.

**Figure 13 sensors-22-04054-f013:**
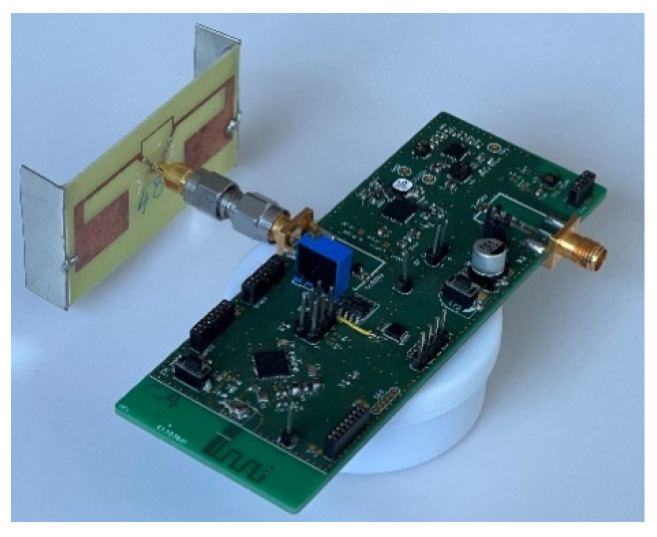
Sensing node connected to an optimized antenna.

**Figure 14 sensors-22-04054-f014:**
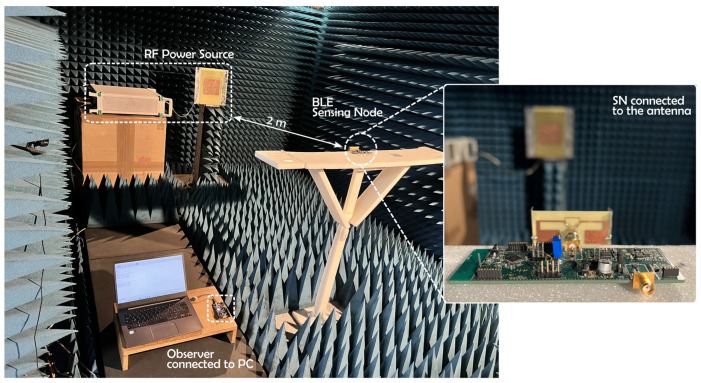
Measurement setup of the charge duration of the SN connected to an antenna in an anechoic chamber.

**Figure 15 sensors-22-04054-f015:**
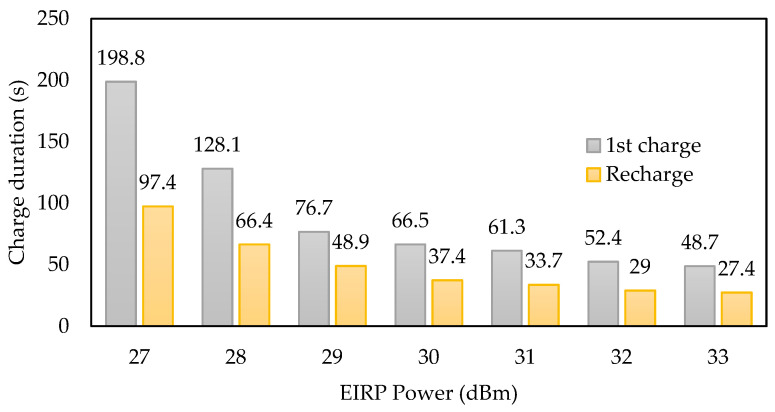
Measured charge duration of the SN with a connected antenna in an anechoic chamber with different EIRP power levels.

**Table 1 sensors-22-04054-t001:** State of the art of several wireless sensors powered by different energy harvesting techniques.

Ref.	EH Sources	Power Density	Applications	Communicating Protocol	Size	Harvested Power
[19]	Solar: Indoor Outdoor	600 lux n. a	Wearable safety (CO_2_, T° and H)	LoRaWAN	Round solar panel (radius of 30 mm)	0.7 mW (@1.24 V) 90 mW (@1.8 V)
[20]	Piezoelectric	600 με at 10 Hz	Aircraft (T°, H and Acc.)	ZigBee	50 mm × 85 mm	3.2 mW
[12]	Thermal	110 °C with a heater	Industrial plants (T°)	ZigBee	40 mm × 40 mm	3.6 mW
[21]	RF	+26 dBm at 868 MHz	IoT	BLE	60 mm × 40 mm	48 µW (at 4 m)

Monitored parameters: T° is temperature, H is humidity, Acc. is accelerometer.

**Table 2 sensors-22-04054-t002:** Comparison of the key parameters of most used BLE transceivers in IoT applications.

Parameters	QN908x [30]	QN9090 [31]	nRF52833 [32]	BlueNRG-LP [33]	RSL10 [34]	EFR32BG22 [35]	DA14531 [36]
Sensitivity (dBm) ^1^	−95	−97	−95	−97	−94	−98.9	−94
Tx power (dBm)	−30 to +2	Up to +11	−20 to +8	−20 to +8	−17 to +6	−27 to +6	−19.5 to +2.5
Rx current (mA)	3.5	4.3	6.0	3.4	3.0	3.6	2.2
Tx current (mA) ^2^	3.5	7.4	6.0	4.3	4.6	4.1	3.5
Deep-sleep mode current (nA)	1000	350	1300	900	100	1050	1200
Supply voltage (V)	1.62 to 3.6	1.9 to 3.6	1.7 to 5.5	1.7 to 3.6	1.1 to 3.3	1.71 to 3.8	1.1 to 3.3
ADC	16-bit 8-channel	12-bit 8-channel	12-bit	12-bit 8-channel	8 to 14-bit	12-bit	10-bit

^1^ The sensitivity is specified in 1 Mbps mode. ^2^ Value obtained for 0 dBm TX power with DC-DC.

**Table 3 sensors-22-04054-t003:** Component value of the rectifier.

Frequency	L1	C1
868 MHz	33 nH LQW15AN33NG00	4 pF GRM1551X1H4R0CA01D
2.45 GHz	3.9 nH LQW15AN3N9B00D	2.1 pF GRM1553C1H2R1BA01D

**Table 4 sensors-22-04054-t004:** Current consumption during each state of a broadcasting event.

	State Description	Time Duration (ms)	I (µA)
(4)	Deep-sleep mode: Between advertising events	234	31.4
(2)	First state (start-up, init., cal. and 1st adv. Event)	241	1074
(3)	A full advertising event	8	1758
(5)	Offset cancelation and stop advertising	11	1032
	Total broadcasting event (from start-up to stop adv.)	1216	282

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
