# Peer review of "A Multifunctional Battery-Free Bluetooth Low Energy Wireless Sensor Node Remotely Powered by Electromagnetic Wireless Power Transfer in Far-Field"

_sensors, 2022, doi:10.3390/s22114054_

Round 1
Reviewer 1 Report
In this paper, the authors present a multifunctional battery-free wireless Sensing Node (SN), part of a Wireless Sensor Network (WSN), designed to monitor the physical parameters of reinforced concrete. The authors present the architecture of the proposed system which consists of a rectifier, specifically designed to convert the Radiofrequency (RF) power to dc power energy and supply a wireless SN; two Power Management units (PMU) to efficiently manage and store the required energy in a storage capacitor; sensors and a transceiver that allow wireless transmission of the measured temperature and humidity data.
The paper is well written, authors may consider the following comments to update the quality of the paper.
- Contributions of the work are not perfectly highlighted in the abstract and Introduction. My suggestion is to rewrite the abstract and highlight the main contribution of the paper in the Introduction section.
- Please enlarge Fig. 1 to make the readability of the paper. Text inside the box is not clear to read.
- According to the motivation of the work, it’s not clear about sensor nodes networks. Is sensor nodes form in a cluster or flat network. Normally, the sensor node used the clustering method for communication and transfer data. Please make sure type of network in your application scenario.
- Authors missed some recent work in WPSN area. Following is some example
https://www.mdpi.com/1424-8220/22/8/2952
https://ieeexplore.ieee.org/document/9722365
https://ieeexplore.ieee.org/document/9750980
Author Response
Thank you for accepting the review of our paper. The latest version of the manuscript includes the suggestions.
1- The abstract and introduction are rewritten.
2- Figure 1 is removed to keep the topic of this paper on the sensor node.
3- The paper deals with a specific sensor node, not a sensor network, but the aim is to get multiple nodes and achieve the expected performance. The architecture of the system is based on nodes forming a cluster network. More details are reported in references 7 and 8.
4- Interesting papers, thank you for sharing. They are included in the papers
Reviewer 2 Report
About the title: I believe that the title is misleading. After reading the paper, I believe it deals with a Wireless Sensor Node rather than a Wireless Sensor Network.
The paper deals with an well-known field of RF WPT that has already been treated in the past by several other authors including:
1) Castorina, G.; Di Donato, L.; Morabito, A.F.; Isernia, T.; Sorbello, G. Analysis and design of a concrete embedded antenna for wireless monitoring applications [antenna applications corner]. IEEE Antennas Propag. Mag. 2016, 58, 76–93.
2) La Rosa, Roberto, et al. "Strategies and techniques for powering wireless sensor nodes through energy harvesting and wireless power transfer." Sensors 19.12 (2019): 2660.
3) La Rosa, Roberto, Catherine Dehollain, and Patrizia Livreri. "Advanced monitoring systems based on battery-less asset tracking modules energized through rf wireless power transfer." Sensors 20.11 (2020): 3020.
For this reason I cannot give a high rate regarding the novelty aspect of the paper. However I can see that the implementation has been carried with method and accuracy.
I would suggest the following improvements:
1) Line 24: Rf-to-dc --> RF-to-DC.
2) Line 51: chapter 2 or section 2 ?
3) Line 123-130: the authors state that RFID has the disadvantage of short-range communication. I would suggest the author to quantify with numbers the communication range. I believe this point should be further investigated by the authors. I have seen in the article a very brief discussion about the distance range from which the WPT is performed from line 134 to line 137. I would invite the authors to be more specific. For instance 11 meters? --> What is the power that is transmitted from the CN? Is there any regulation that it must be respected? etc.
11 meters and 3 meters --> in which conditions? At which frequency? 868 MHz or 2.4 Ghz?
I believe that this article is worth of an improvement of this topic that to me is the most interesting if I aim to use this system in a real use case.
4) Further, I have not seen a solid explanation in the paper of why RFID is preferable to the BLE communication. I can see that at 868 MHz the power sensitivity of the rectifier is -14 dbm, and with the allowed power (in Europe) of 27 dBm the range is about 3 meters. These performances are quite aligned to that of an RFID system in terms of distance. In Figure 14, I see a distance of 2 meters. That said, I believe that a deeper discussion on the convenience of using a beacon over the RFID should be worth this article.
5) Line 137-138: "In this work, a tradeoff has been made between the range and the power con- 137 sumption by choosing BLE technology".
I would be happy to read with numbers about the tradeoff.
6) Table 2 and BLE System on Chip: I believe that comparison between the different BLE SoC based on table 2 is not enough. I would encourage the Authors to make a comparison among the diferent SoC base don the data from the calulator that each vendor provides. These tool are more accurate and include more details. Therefore I would suggest to base the comparison on the basis of Equation (1) but analyzing each of the SoC mentioned in the article.
7) Line 185-187: I do not understand this passage. The article is mentioning a previous work that is not referenced. As it is I have difficulties to understand the added scientific value of this sentence.
8) Line 198-201: The abstract mentions that the wireless sensor node aim to sense temperature and humidity. According to what is written in line 198-201 the system can sense also corrosion rate. This is an interesting feature. Therefore, I would suggest the authors to mention this in the abstract and expand this part of the article with further details.
9) Line 226-228: "SMS7630 is selected de to its low biasing voltage"
How low? Low compared to what and who?
10) Line 235: I believe that table 1 should be replaced with table 3.
11) Equation (2) should be explicitated in terms of C. It would be nice to have a table for the Econs of the different SoC mentioned calculated as for the QN9080. This would sound more like a fair apple-to-apple comparison for energy performance.
12) Line 363: what is the power used?
13) Figure 15 is a nice graph. I would suggest to make a similar graph with in the y axis the distance between the Cn and the SN.
Author Response
Thank you for accepting the review of our paper. The latest version of the manuscript includes the suggestions. The changes made are highlighted in blue.
You are right, this paper deal with a sensor node, results of the global network is not presented but it is the aim of the project. I will ask the journal editor to change the title (replace Network by Node).
1- Checked
2- Checked
3- Checked
4- Checked
5- Reformulated
6- Not all SoCs allow this functionality. The use of the calculation tool requires the programming of each SoC (we do not find it useful). Equation 1 cannot be applied to other SoCs (not used in this work). Estimating the current during each state is not possible from the vendor information.
7- Checked
8- The information has been added in the abstract but experimentations are not available yet for this part.
9- In the state of the art is studied the performance of different schottky diode in the field of energy harvesting. More details have been given in the paper.
10- Checked
11- Econs, is the overall energy consumption of the SN (sensors and QN9080). This evaluation can only be performed when the sensor and the BLE transceiver are powered. The configuration of the QN9080n does not allow broadcasting without any data from the sensor.
12- The measurements were obtained at different power levels.
13- The distance between the CN and the SN has not been evaluated. Regarding the low sensitivity of the BLE transceiver, a coverage of more than 10 m can be expected.
Reviewer 3 Report
The paper presents a battery-free wireless sensor node which is designed for monitoring some physical attributes of reinforced concrete. The sensor node is intended to be embedded into the concrete cavity and is wirelessly powered via the novel technique of wireless power transfer. The sensor node has been implemented, and some of its performance data are given and discussed.
The presented work is not theoretical but it demonstrates a good implementation example of the stare-of-the-art techniques in wirelessly powered sensors and networks. The practical result of implementation and real measurement may be very useful to researchers and engineers in the domain of wirelessly powered sensors and networks.
It is much better if the authors present some more measurement results including performance data additionally. Also, it is required to discuss the shortcomings and limitations of the implemented sensors for readers in the same domain.
It is necessarily required to include some recent works on wireless power transfer in sensors and sensor networks, which have been published in major conference proceedings and reputed journals including Sensors in 2022.
Author Response
Thank you for accepting the review of our paper. The latest version of the manuscript includes the suggestions. The changes made are highlighted in blue.
Some recent works on wireless power transfer in sensors and sensor networks are included in the version of the submitted manuscript.
Round 2
Reviewer 1 Report
Thanks for revision.
Author's address all my initial comments on the paper. I think paper can be accepted as current condition.
Reviewer 2 Report
This improved version is satisfactory to me.
I would only suggest the evaluation of the following paper as a reference in the context of the wireless power transfer.
"La Rosa, Roberto, Catherine Dehollain, and Patrizia Livreri. "Advanced monitoring systems based on battery-less asset tracking modules energized through rf wireless power transfer." Sensors 20.11 (2020): 3020."